# Examination of Surfactant Protein D as a Biomarker for Evaluating Pulmonary Toxicity of Nanomaterials in Rat

**DOI:** 10.3390/ijms22094635

**Published:** 2021-04-28

**Authors:** Taisuke Tomonaga, Hiroto Izumi, Yukiko Yoshiura, Chinatsu Nishida, Kazuhiro Yatera, Yasuo Morimoto

**Affiliations:** 1Institute of Industrial Ecological Sciences, University of Occupational and Environmental Health, Kitakyusyu 807-8555, Fukuoka, Japan; h-izumi@med.uoeh-u.ac.jp (H.I.); y-yoshiura@med.uoeh-u.ac.jp (Y.Y.); yasuom@med.uoeh-u.ac.jp (Y.M.); 2Department of Respiratory Medicine, University of Occupational and Environmental Health, Kitakyusyu 807-8555, Fukuoka, Japan; c-nishi@med.uoeh-u.ac.jp (C.N.); yatera@med.uoeh-u.ac.jp (K.Y.)

**Keywords:** surfactant protein D, kinetics, pulmonary toxicity, biomarker, nanomaterials

## Abstract

This work studies the relationship between lung inflammation caused by nanomaterials and surfactant protein D (SP-D) kinetics and investigates whether SP-D can be a biomarker of the pulmonary toxicity of nanomaterials. Nanomaterials of nickel oxide and cerium dioxide were classified as having high toxicity, nanomaterials of two types of titanium dioxides and zinc oxide were classified as having low toxicity, and rat biological samples obtained from 3 days to 6 months after intratracheal instillation of those nanomaterials and micron-particles of crystalline silica were used. There were different tendencies of increase between the high- and low-toxicity materials in the concentration of SP-D in bronchoalveolar-lavage fluid (BALF) and serum and in the expression of the SP-D gene in the lung tissue. An analysis of the receiver operating characteristics for the toxicity of the nanomaterials by SP-D in BALF and serum showed a high accuracy of discrimination from 1 week to 3 or 6 months after exposure. These data suggest that the differences in the expression of SP-D in BALF and serum depended on the level of lung inflammation caused by the nanomaterials and that SP-D can be biomarkers for evaluating the pulmonary toxicity of nanomaterials.

## 1. Introduction

New nanomaterials are being developed day after day accompanying recent technological innovations and are used in various industrial fields. These nanosized materials have new physicochemical properties and are used not only in industrial applications such as semiconductors but also in daily necessities such as cosmetics and sunscreens. At the same time, there is concern that the various physicochemical properties of nanomaterials can have harmful biological effects. It has been reported that nano-sized particles induce more severe inflammation in the lung than micron-sized particles [1,2]; therefore, continuous assessment of the hazards of newly developed nanomaterials is necessary for their safe use.

It is known that inhalable dust that is taken into the lung and deposited there causes lung disorders. Silica and asbestos are known to cause persistent inflammation resulting in irreversible fibrosis and tumors due to long-term deposition in the lung [3,4,5]. On the other hand, it has been reported that titanium dioxide nanoparticles and fullerene nanoparticles, which are considered to be low in pulmonary toxicity, cause transient lung inflammation but not progression to fibrosis or tumor in the lung [6,7]. Alveolar epithelial cells in the lung are the first line of defense in the exposure to inhalable dust that is taken into the lung. Among them, type II alveolar epithelial cells play a role in the secretion of pulmonary surfactant, which is responsible for maintaining lung compliance [8,9]. Since surfactant protein D (SP-D) is a surfactant protein and is mainly produced by type Ⅱ alveolar epithelial cells and club cells which are present in the bronchial and alveolar epithelium [8,10], it can be used for clinical diagnosis and evaluation of the progress of the diseases of interstitial pneumonia and pulmonary fibrosis, which induce epithelial injury in the lung [11,12,13].

SP-D expression has been observed in alveolar epithelial injury models in animal studies. It has been reported that the concentration of SP-D in bronchoalveolar-lavage fluid (BALF) and serum increases due to lung inflammation caused by nanomaterials [14,15], and it is considered that it could be a marker for evaluating the pulmonary toxicity of nanomaterials. In order to verify the usefulness of SP-D as a marker for evaluating the pulmonary toxicity of nanomaterials, it is important to examine the kinetics of SP-D in lung inflammation caused by nanomaterials, but there are no reports focusing on the kinetics of SP-D in exposure to nanomaterials, and there is not enough data to examine SP-D as a useful marker for evaluating their pulmonary toxicity. Therefore, in the present study, in order to clarify the relationship between lung inflammation caused by nanomaterials and SP-D kinetics, we measured SP-D in BALF, lung tissue and blood samples obtained from intratracheal instillation of nanomaterials with different pulmonary toxicities in rats and analyzed the relationship between SP-D and inflammatory markers in lung inflammation caused by nanomaterials. Based on SP-D kinetics, we investigated whether SP-D could be a useful biomarker for evaluating the pulmonary toxicity of nanomaterials.

## 2. Results

### 2.1. Pathological Features in the Rat Lung

Figure 1 shows the pathological findings in the rat lung at 6 months after the intratracheal instillation of the nanomaterials. In the intratracheal instillation of nanoparticles of nickel oxide (NiO), cerium dioxide (CeO_2_) and micron-particles of crystalline silica (SiO_2_), there was infiltration of inflammatory cells such as neutrophils and macrophages in alveolar spaces in the lung at 6 months after exposure (Figure 1A–C). In the intratracheal instillation of two types of titanium dioxide nanoparticles (TiO_2_ (P90) and TiO_2_ (rutile)), zinc oxide nanoparticles (ZnO) and distilled water as a negative control, there was no infiltration of inflammatory cells or fibrosis at 6 months after exposure (Figure 1D–G). 

### 2.2. SP-D Concentrations in BALF

Figure 2 shows the SP-D concentration in the BALF at each time point after intratracheal instillation of NiO, CeO_2_, TiO_2_ (P90), TiO_2_ (rutile), ZnO and SiO_2_. Increases in the SP-D concentration in BALF were observed until 1 month in the groups with a low-dose (0.2 mg/rat) exposure to nanomaterials with high toxicity (NiO and CeO_2_) (Figure 2A). In the high dose (1.0 mg/rat) exposures, the SP-D concentration in BALF increased persistently and in a dose-dependent manner compared to the low-dose exposures (Figure 2B). In the exposure to nanomaterials with low toxicity (TiO_2_ (P90), TiO_2_ (rutile) and ZnO), the SP-D concentration in BALF increased mainly at 3 days and 1 week. Even in the high-dose (1.0 mg/rat) exposure, there was a tendency of only transient increase in the SP-D concentration in BALF. 

### 2.3. Correlation between SP-D and Inflammatory Markers in BALF

In order to examine the SP-D kinetics in lung inflammation caused by nanomaterials, the relationship between SP-D and inflammatory markers was analyzed. Figure 3 shows the correlation between the SP-D concentration and inflammatory markers in BALF. The SP-D concentration in the BALF correlated well with the neutrophil count, the total cell counts, the rat heme oxygenase (HO)-1 concentration as an oxidative stress marker, cytokine-induced neutrophil chemoattractant (CINC)-1, CINC-2 as chemokines and the activity of released lactate dehydrogenase (LDH) as a lung injury marker.

### 2.4. Gene Expression Analysis in Lung Tissue

Figure 4 shows the validated expression levels of the SP-D genes induced by the five nanomaterials and the micron-particles of SiO_2_ following intratracheal instillation using qRT-PCR over the observation time. The gene expression of SP-D in the lung tissue exposed to NiO, CeO_2_ and SiO_2_, which have high pulmonary toxicity, was persistently high compared with the negative control throughout the observation time, while, on the other hand, the expression increased transiently in the lung tissue exposed to TiO_2_ (P90), TiO_2_ (rutile) and ZnO, which have low pulmonary toxicity. 

### 2.5. Immunostaining of SP-D in Lung

Figure 5 shows the immunostaining of SP-D in the lung at 3 months after exposure to the high dose of NiO and the negative control. In the lung exposed to NiO, positive staining of SP-D was observed not only in type II alveolar epithelial cells but also in alveolar macrophages and alveolar mucus.

### 2.6. SP-D Concentrations in Serum

Figure 6 shows the SP-D concentration in the serum at each time point after intratracheal instillation of NiO, TiO_2_ (P90) and SiO_2_. Significant increases in the SP-D concentration in serum compared to each negative control were observed in the chronic phase in the groups of high-dose (1.0 mg/rat) exposure to NiO and SiO_2_, while there was no significant increase in SP-D concentrations in serum in the groups of TiO_2_ (P90) exposure. 

### 2.7. The Relationship between SP-D Concentration in BALF and Serum

Figure 7A shows the relationship between the SP-D concentrations in BALF and in serum from each sample. The Spearman correlation coefficient was 0.439 (*p*-value: 0.000). There was a weak correlation between the SP-D concentrations in the BALF and in the serum. Figure 7B–D shows the relationship between the SP-D concentrations in serum and inflammatory markers from each sample. SP-D in serum was correlated with inflammatory cells and total proteins in lung inflammation caused by exposure to nanomaterials. Table 1 shows a summary of the SP-D concentration in BALF and serum at 3 months after intratracheal instillation of chemicals including nanomaterials for comparison with the changes that occur during inflammation. In the exposure to nanomaterials with high toxicity, the rate of increase in SP-D in BALF and serum (exposure/control ratio) ranged from 4-fold to 7-fold, and from 1.2-fold to 1.6-fold, respectively. The maximum rate of increase in SP-D in BALF was 11.2-fold at 1 week after exposure to the high dose of CeO_2_ (data not shown). These results indicate a slight increase in SP-D concentration in serum compared to that in BALF.

### 2.8. Assessment of the Accuracy of SP-D in BALF and Serum for Measuring the Toxicity of Nanomaterials

Figure 8 shows the results of the receiver operating characteristic (ROC) for the toxicity of the nanomaterials by the SP-D concentration in BALF and serum after the intratracheal instillation. There were statistically significant AUCs of SP-D in BALF and serum from 1 week to 6 months and from 1 week to 3 months after exposure, respectively. In order to examine the accuracy of SP-D for evaluating pulmonary toxicity of nanomaterials, areas under the curve (AUCs) were analyzed. The largest AUC using ROC curves for the toxicity of the nanomaterials in SP-D in BALF was 0.982 (95% CI, 0.952–1.000) at 1 month. The largest AUC in SP-D in serum was 1.000 (95% CI, 1.000–1.000) at 1 month after exposure as well (Table 2). 

## 3. Discussion

In the present study, we analyzed SP-D concentration in BALF and serum and SP-D mRNA expression in lung tissue to investigate the kinetics of SP-D due to exposure to nanomaterials. First, we analyzed the kinetics of SP-D in exposure to NiO nanoparticles, CeO_2_ nanoparticles and micron-sized particles of SiO_2_ which had high pulmonary toxicity. SP-D concentration in BALF increased significantly in SiO_2_, NiO and CeO_2_ at 3 days to 6 months after exposure. An increase in SP-D concentration in BALF has also been reported in LPS, bleomycin and pneumocystis infection, all of which induce lung inflammation [16,17,18]. LPS and bleomycin have been shown to increase SP-D with acute epithelial injury [16,17,18]. On the other hand, in a chronic inflammation model that used transgenic mouse that overexpresses tumor necrosis factor (TNF)-alpha under the control of the SP-C promoter, an increase in SP-D concentration in BALF was observed in the chronic phase [19]. It is considered that the persistent increase in SP-D concentration in the present study was caused by persistent inflammation and the ensuing lung injury. This view is also supported by the results of the correlation between SP-D concentration in BALF and inflammatory factors. In the present study, the SP-D concentration in BALF correlated not only with the activity of the LDH in BALF, a lung injury marker, but also with the neutrophil counts in BALF, the concentrations of CINCs in BALF as inflammatory cytokines and the HO-1 concentration in BALF as an oxidative stress marker. There was also a persistent increase in SP-D gene expression in lung tissue exposed to SiO_2_, NiO and CeO_2_ at 3 days to 6 months, suggesting that SP-D in BALF was produced in type Ⅱ alveolar epithelial cells and club cells by exposure to nanomaterials. These results suggest that the increase in SP-D concentration in BALF was upregulated due to release from destruction of the epithelial cells and production in type Ⅱ epithelial cells and club cells stimulated by the nanomaterials. 

Immunostaining for SP-D also showed positive staining of SP-D in type Ⅱ alveolar epithelial cells and alveolar macrophages due to NiO exposure. The positive staining of SP-D in the macrophages is considered to be due to the macrophages phagocytosing SP-D that was released into the alveolar space.

Since SP-D in serum can detect and is also used as a biomarker of disease progression for interstitial pneumonia [13,20], we examined the SP-D concentration in serum due to exposure to nanomaterials. In general, since SP-D is mainly produced in type Ⅱ alveolar epithelial cells and club cells [8,10], SP-D in serum is considered to be derived from the lung. Although the exact mechanism of the increase in SP-D in serum associated with lung injury has not been elucidated so far, it is thought that its cause is the destruction of the vascular endothelium as a blood–air barrier and/or an increase in vascular permeability due to lung inflammation [16,17]. In the present study, a significant increase in SP-D concentration in serum was observed only in the chronic phase of 3 and 6 months after exposure, and the time to the peak in the increase in the concentration of SP-D in serum was delayed compared to that in BALF. We previously observed a persistent increase in total protein in BALF, which reflects vascular permeability, in the same exposure to NiO and CeO_2_ as in the present study [21]. Furthermore, in the present study, SP-D in serum was correlated with inflammatory cells and total protein in BALF. Although SP-D in serum had a weaker correlation with inflammatory cells in BALF than SP-D in BALF, the reason for this may be that there was a temporal difference in the increase in SP-D in serum. Persistent inflammation due to exposure to nanomaterials may have gradually enhanced the vascular permeability, resulting in significant detection of SP-D in serum in the chronic phase. On the other hand, we observed from the results of immunostaining a notably large number of macrophages in the alveolar space that appeared to have phagocytized SP-D. It was considered that the transition of SP-D to the blood may have been mild due to the macrophages having metabolized SP-D in the alveolar space.

Regarding the nanomaterials with low pulmonary toxicity, there was an increase in SP-D concentration in BALF and SP-D gene expression in the lung tissue, but the changes were transient. As a result, there was no increase in SP-D concentration in the serum in the exposure to the nanomaterials with low toxicity. The reason was that there was not enough production of SP-D in the lung to transfer to the blood or that there could probably have been an absence of persistent increase in vascular permeability since the protein concentration in BALF did not show a persistent increase in our previous analysis of the exposure to nanomaterials with low pulmonary toxicity [21].

Based on SP-D kinetics, we examined whether SP-D can be useful as a marker for evaluating the pulmonary toxicity of nanomaterials. In the same way as in previous studies [22,23], we classified NiO and CeO_2_, which cause persistent lung inflammation, as having high pulmonary toxicity, and TiO_2_ (P90), TiO_2_ (rutile) and ZnO, which cause transient lung inflammation, as having low pulmonary toxicity. We examined the sensitivity and specificity of SP-D concentration in BALF and serum as biomarkers for the evaluation of pulmonary toxicity by conducting an ROC analysis in the intratracheal instillation of nanomaterials (Figure 8, Table 2) and found a significant value of AUC in SP-D concentration in BALF from 1 week to 6 months after exposure. The results of the ROC analysis were interpreted as follows: AUC < 0.70, low diagnostic accuracy; AUC in the range of 0.70–0.90, moderate diagnostic accuracy; and AUC ≥ 0.90, high diagnostic accuracy [24]. We surmise that SP-D concentration in BALF at 1 month could most accurately reflect the rank of pulmonary toxicity. Even though nanomaterials such as TiO_2_ (P90), TiO_2_ (rutile) and ZnO have low pulmonary toxicity, injury of the alveolar epithelium involved in lung inflammation and enhancement of SP-D production were observed in the acute phase. Therefore, for evaluation of the pulmonary toxicity of nanomaterials, it is considered to be useful to examine the SP-D concentration in BALF at the subacute phase, when lung inflammation caused by nanomaterials with low toxicity has recovered, such as at 1 month after exposure.

Regarding SP-D in serum, there were significant values of AUC in SP-D concentration in serum at 1 week, 1 month and 3 months after exposure in the ROC analysis of NiO and TiO_2_ (P90), and SP-D concentration in serum at 1 month could most accurately reflect the rank of pulmonary toxicity. From the kinetics of SP-D in serum, considering that persistent lung inflammation enhances the production of SP-D in the lung and the transfer of SP-D from the lung to the blood, in order to evaluate the pulmonary toxicity of nanomaterials, it would be useful to examine SP-D in serum up to about 3 months after exposure to nanomaterials, when persistent lung inflammation is sufficiently induced. 

In the present study, AUCs of SP-D in BALF and serum were shown around 1 in ROC analysis and had high accuracy for judgment of pulmonary toxicity of nanomaterials. Since nanomaterials that were able to be clearly divided into high- and low-toxicity were used in ROC analysis in order to explore a useful biomarker for evaluating pulmonary toxicity of nanomaterials, AUCs were thought to be shown around 1. In the future, it is necessary to include nanomaterials that have borderline toxicity and to take a further examination that SP-D could be a useful biomarker for evaluating pulmonary toxicity of nanomaterials.

## 4. Materials and Methods

### 4.1. Sample Nanomaterials

NiO (US3355, US Research Nanomaterials, Houston, TX, USA), two types of TiO_2_ (P90) (AEROXIDE Evonik Degussa Corp, Nordrhein-Westfalen, Germany) and TiO_2_ (rutile) (MT-150 AW, Teyca Co. Ltd., Osaka, Japan), CeO_2_ (Wako Chemical, Ltd., Osaka, Japan), ZnO (Sigma-Aldrich Co. LLC., Tokyo, Japan) and micron-particles of SiO_2_ (MIN-U-SIL^®^ 5, U.S. Silica Company, Houston, TX, USA) were each dispersed in 0.4 mL distilled water. 

The physicochemical profiles of these samples are shown in Table 3. The data of these samples have been partially published in previous studies [25,26,27,28,29]. As in our previous study [22,23], we defined the toxicity of the chemicals as follows: the chemicals that induced either persistent inflammation, fibrosis or tumor in intratracheal instillation studies were defined as having high toxicity, and the chemicals that did not induce any of these pathological conditions were defined as having low toxicity. Accordingly, NiO and CeO_2_ were classified as chemicals with high toxicity, and TiO_2_ (P90), TiO_2_ (rutile) and ZnO were classified as chemicals with low toxicity.

### 4.2. Animals

Male Fischer 344 rats (9–11 weeks old) used in the exposure to NiO, CeO_2_, TiO_2_ (rutile), ZnO and SiO_2_ were purchased from Charles River Laboratories International, Inc. (Kanagawa, Japan). Male Wistar Hannover rats (11 weeks old) used in the exposure to TiO_2_ (P90) were purchased from Japan SLC, Inc. (Shizuoka, Japan). The animals were kept in the Laboratory Animal Research Center of the University of Occupational and Environmental Health for 2 weeks with access to free-feeding of commercial diet and water. All procedures and animal handling were done according to the guidelines described in the Japanese Guide for the Care and Use of Laboratory Animals as approved by the Animal Care and Use Committee, University of Occupational and Environmental Health, Japan (approval number: AE11-012).

### 4.3. Intratracheal Instillation

The NiO, TiO_2_ (P90), TiO_2_ (rutile), CeO_2_, ZnO and SiO_2_ were suspended in 0.4 mL distilled water, and 0.2 mg (low dose) or 1 mg (high dose) was administered to rats (12 weeks old) in single intratracheal instillations. SiO_2_ was used as a positive control. Each of the negative control groups received dispersion mediums that used the suspensions in each exposure examination. Animals were dissected at 3 days, 1 week, 1 month, 3 months and 6 months after the instillation.

### 4.4. Animals Following Intratracheal Instillation

There were 10 rats in each group of exposure to NiO, TiO_2_ (P90), TiO_2_ (rutile), CeO_2_, ZnO, SiO_2_ and the control, divided into two subgroups of five animals in each low-dose and high-dose group at each time point. In each of the first subgroups, five rats provided bronchoalveolar lavage at each time point. The lungs were inflated with 20 mL of saline at a water pressure of 20 cm, and BALF was collected from the entire lung in two or three portions. Fifteen to eighteen milliliters of BALF was collected in a collection tube by free fall. Serum samples of NiO, TiO_2_ (P90) and SiO_2_ were collected by cardiopuncture. In the second subgroup, the lungs were divided into right and left lungs. Analysis of qRT-PCR was performed with the homogenized third lobe of the right lung, and histopathological evaluation was performed with the left lung inflated and fixed by 10% formaldehyde. 

### 4.5. Analysis of Inflammatory Cells in BALF with Cytospin

The obtained BALF was centrifuged at 400× *g* at 4 °C for 15 min, and the supernatant was transferred to a new tube and frozen for measuring the cytokines. The pellets were washed by suspension with polymorphonuclear leukocyte (PMN) buffer (137.9 mM NaCl, 2.7 mM KCl, 8.2 mM Na_2_HPO_4_, 1.5 mM KH_2_PO_4_, 5.6 mM C_6_H_12_O_6_) and centrifuged at 400× *g* at 4 °C for 15 min. After the supernatant was removed, the pellets were re-suspended with 1 mL of PMN buffer. The number of cells in the BALF was counted by Celltac (Nihon Kohden Corp., Tokyo, Japan), and the cells were splashed on a slide glass using cytospin. After the cells were fixed and stained with Diff-Quik (Sysmex Corp., Hyogo, Japan), the number of neutrophils and alveolar macrophages was counted by microscopic observation. These data have been published in our previous reports [22,23,25,26,27,28,29].

### 4.6. Measurement of SP-D, Chemokines, Lactate Dehydrogenase and Heme Oxygenase-1 in BALF

The concentrations of rat SP-D in the BALF samples from all of the examinations and in the serum samples of NiO, TiO_2_ (P90) and SiO_2_ were measured by an ELISA kit (Yamasa Corporation, Chiba, Japan). The concentrations of rat CINC-1 and CINC-2 in the BALF samples from all of the examinations were measured by ELISA kits #RCN100 and #RCN200 (R&D Systems, Minneapolis, MN, USA), respectively. The concentration of rat HO-1 in the NiO, TiO_2_ (P90), TiO_2_ (rutile), CeO_2_ and ZnO exposure examinations were measured by an ELISA kit ADI-EKS-810A (Enzo Life Sciences, Farmingdale, NY, USA). The activity of released LDH in the NiO, TiO_2_ (rutile), CeO_2_ and ZnO exposure examinations was measured by a Cytotoxicity Detection KitPLUS (LDH) (Roche Diagnostics GmbH, Mannheim, Germany). All measurements were performed according to the manufacturer’s instructions. These data, except for the SP-D, have been published in our previous reports [22,23,25,26,27,28,29].

### 4.7. Total RNA Extraction

The third lobes of the right lungs (n = 5 per group per time point) were homogenized with a TissueRuptor (Qiagen, Hilden, Germany). Total RNA from the homogenates was extracted using an RNeasy Mini Kit (Qiagen, Hilden, Germany) following the manufacturer’s instructions. RNA was quantified using a NanoDrop 2000 spectrophotometer (Thermo Fisher Scientific Inc., Waltham, MA, USA).

### 4.8. Validation of Gene Expression Data Using Quantitative Real-Time Polymerase Chain Reaction

qRT-PCR was performed as described previously [30,31]. Briefly, the total RNA extracted from the lungs at each observation point in each group was transcribed into cDNA (High-Capacity cDNA Reverse Transcription Kit, Thermo Fisher Scientific Inc., Waltham, MA, USA). qRT-PCR assays were performed while using TaqMan (TaqMan Gene Expression Assays, Thermo Fisher Scientific Inc., Waltham, MA, USA) according to the manufacturer’s protocol. Gene expression data were analyzed by the comparative cycle time (ΔΔCT) method. The Assays-on-Demand TaqMan probes and primer pair were SP-D (Assay ID Rn00563557_m1). All experiments were performed in a StepOnePlusTM Real-Time PCR Systems (Thermo Fisher Scientific Inc., Waltham, MA, USA). All expression data were normalized to endogenous control β-actin expression (Assay ID Rn00667869_m1) and calculated relative to gene expression of SP-D in each negative control.

### 4.9. Histopathology

The obtained lung tissue, which was inflated and fixed with 10% formaldehyde or 4% paraformaldehyde under a pressure of 25 cm water, was embedded in paraffin, and 5-µm-thick sections were cut from the lobe, then stained with hematoxylin and eosin. The SP-D of distribution in the inflammation was evaluated by SP-D (sc-7709; Santa Cruz Biotechnologies, Inc., Dallas, CA, USA) immunostaining using lung tissue samples of 3 months after intratracheal instillation in the high dose of NiO group and the negative control.

### 4.10. Statistical Analysis

Analysis of variance and Dunnett’s test were applied where appropriate to determine individual differences using a computer statistical package (SPSS, SPSS Inc., Chicago, IL, USA). Construct validity was measured using Spearman’s rank correlation coefficients between the concentration of SP-D, neutrophil counts, total cell counts, concentration of HO-1, concentration of CINC-1, concentration of CINC-2 and activity of released LDH. We assigned the toxicity of the exposure nanomaterials as being high or low according to the SP-D concentration at each time point in BALF (exposure to NiO, TiO_2_ (P90), TiO_2_ (rutile), CeO_2_ and ZnO) and in serum (exposure to NiO and TiO_2_ (P90)) and analyzed the sensitivity and specificity for high toxicity at each time point to create the ROC curves and AUCs.

## 5. Conclusions

In the present study, we focused on the kinetics of SP-D in exposure to nanomaterials, and SP-D was analyzed in bronchoalveolar lavage fluid, lung tissue and serum samples obtained from intratracheal instillation of nanomaterials with different pulmonary toxicities. SP-D concentration in BALF increased similarly to other lung inflammation models such as bleomycin and LPS, and it was thought that the increase in SP-D concentration in BALF was caused by epithelial injury due to nanomaterial exposure and increases in production from type Ⅱ alveolar epithelial cells and club cells. SP-D concentration in serum was lower than that of SP-D in BALF, and a significant increase in SP-D concentration in serum was observed in the chronic phase in exposure to NiO and SiO_2_, which cause persistent inflammation, compared with each negative control. It was considered that the cause of the increase in SP-D concentration in serum was influenced by the degree of lung inflammation caused by nanomaterials and the phagocytosis of macrophages. From the kinetics of SP-D and ROC analyses in the exposure to nanomaterials with different pulmonary toxicities, it was considered that SP-D concentration in BALF at 1 month and serum at 3 months after exposure can accurately evaluate the pulmonary toxicity of nanomaterials. Taken together, there was a difference in expression of SP-D not only in the lung but also in the blood, depending on the level of lung inflammation caused by nanomaterials, suggesting that SP-D concentrations in both BALF and serum can be biomarkers for evaluating the pulmonary toxicity of nanomaterials.

## Figures and Tables

**Figure 1 ijms-22-04635-f001:**
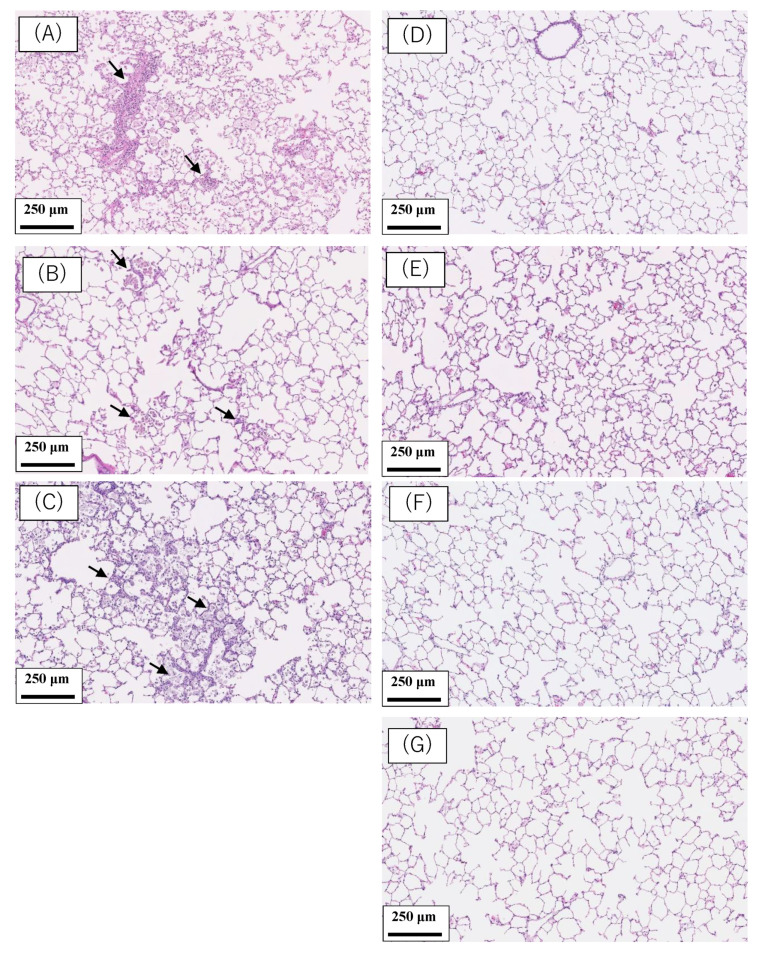
Pathological findings in the rat lung at 6 months after intratracheal instillation of 5 nanomaterials and micron-particles of SiO_2_. (**A**) 1 mg NiO-exposed lung. (**B**) 1 mg CeO_2_-exposed lung. (**C**) 1 mg SiO_2_-exposed lung. (**D**) 1 mg TiO_2_ (P90)-exposed lung. (**E**) 1 mg TiO_2_ (rutile)-exposed lung. (**F**) 1 mg ZnO-exposed lung. (**G**) Negative control in intratracheal instillation (distilled water). Infiltration of inflammatory cells was observed in Figure 1(**A**–**C**) (black arrows).

**Figure 2 ijms-22-04635-f002:**
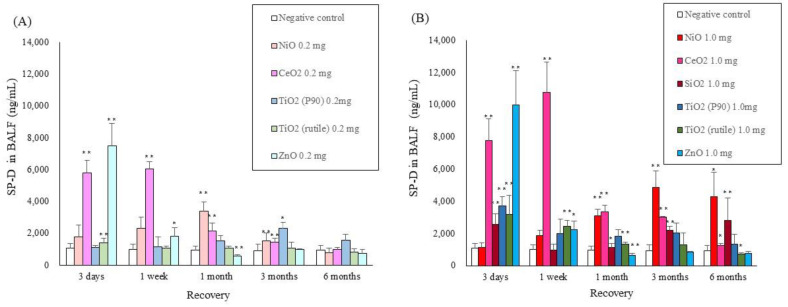
SP-D concentrations in BALF at each time point after intratracheal instillation of nanomaterials: (**A**) SP-D in low dose of intratracheal instillation, (**B**) SP-D in high dose of intratracheal instillation. Error bars represent mean ± SD for n = 5/group. Asterisks indicate significant differences compared with each control (Analysis of variance and Dunnett’s test) (* *p* < 0.05, ** *p* < 0.01). An average of negative controls in all experiments at each time point showed a single bar of negative control.

**Figure 3 ijms-22-04635-f003:**
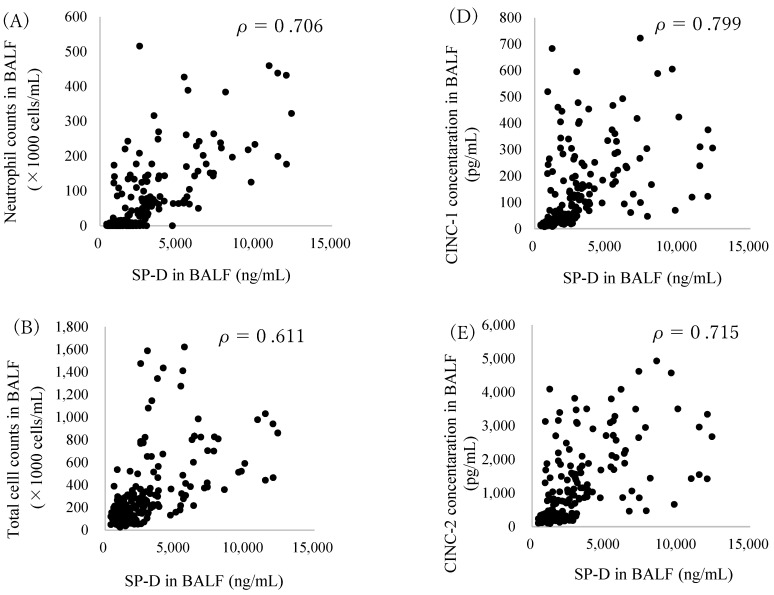
Relationship between SP-D concentration and inflammatory markers in BALF: (**A**) neutrophils (n = 250), (**B**) total cell (n = 250), (**C**) HO-1 (n = 250), (**D**) CINC-1 (n = 250), (**E**) CINC-2 (n = 250) and (**F**) LDH (n = 200) versus SP-D in BALF after intratracheal instillation of nanomaterials. Values of ρ are Spearman’s rank correlation coefficient for all the data.

**Figure 4 ijms-22-04635-f004:**
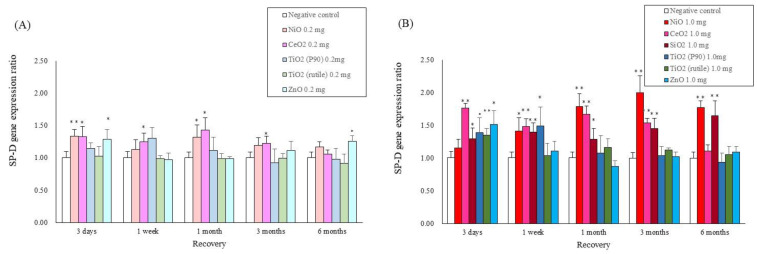
Gene expression of SP-D in lung exposed to nanomaterials with different pulmonary toxicities: (**A**) SP-D gene expression in low dose of intratracheal instillation, (**B**) SP-D gene expression in high dose of intratracheal instillation. Error bars represent mean ± SD for n = 5/group. Asterisks indicate significant differences compared with each control (Analysis of variance and Dunnett’s test) (* *p* < 0.05, ** *p* < 0.01). An average of negative controls in all experiments at each time point showed a single bar of negative control.

**Figure 5 ijms-22-04635-f005:**
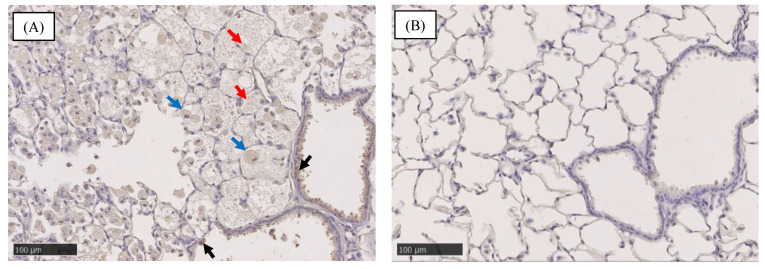
Immunostaining of SP-D in lung at 3 months after exposure. (**A**) 1 mg NiO-exposed lung. (**B**) Negative control in intratracheal instillation (distilled water). Positive staining of SP-D was observed in type II alveolar epithelial cells (black arrow), alveolar macrophages (blue arrow) and alveolar mucus (red arrow) in Figure 5A.

**Figure 6 ijms-22-04635-f006:**
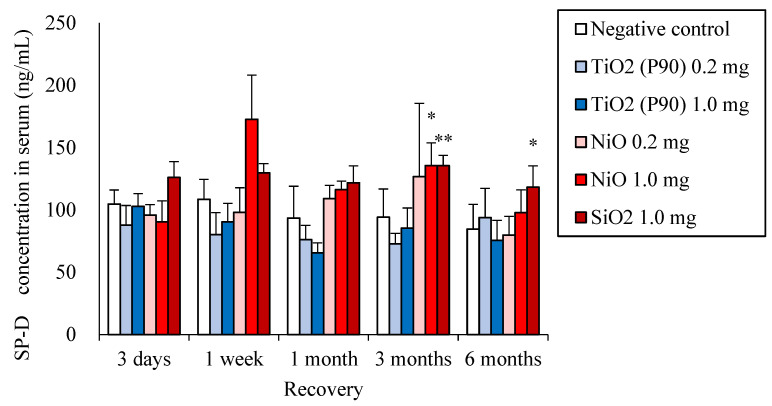
SP-D concentration in serum at each time point after intratracheal instillation of nanomaterials. Error bars represent mean ± SD for n = 4–5/group. Asterisks indicate significant differences compared with each control (Analysis of variance and Dunnett’s test) (*p<0.05, **p<0.01). An average of negative controls in all experiments at each time point showed a single bar of negative control.

**Figure 7 ijms-22-04635-f007:**
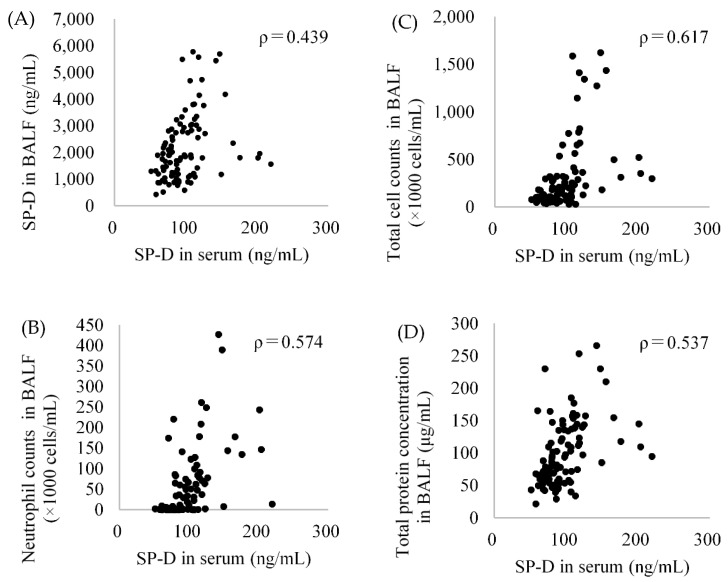
Relationship between SP-D in serum and SP-D in BALF (**A**), neutrophil counts in BALF (**B**), total cell counts in BALF (**C**) and total protein concentration in BALF (**D**) after intratracheal instillation of nanoparticles of NiO and TiO_2_ (P90) (n = 97). Values of ρ are Spearman’s rank correlation coefficient.

**Figure 8 ijms-22-04635-f008:**
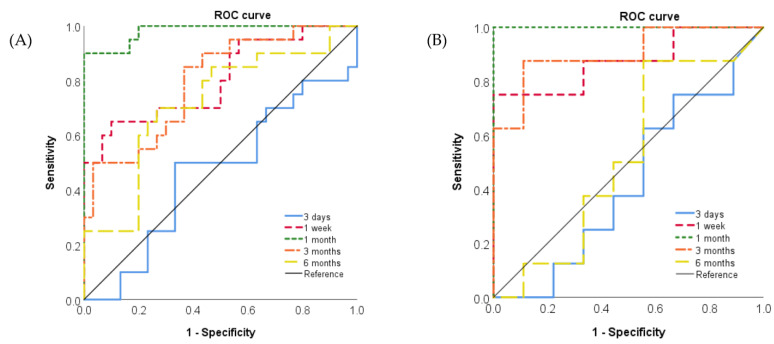
The receiver operating characteristics (ROCs) for the toxicity of nanomaterials by SP-D concentration in BALF and serum. (**A**) ROC curve of SP-D in BALF (n = 250). (**B**) ROC curve of SP-D in serum (n = 97).

**Table 1 ijms-22-04635-t001:** Summary of SP-D concentration in BALF and serum at 3 months after intratracheal instillation of materials.

	Dose	TiO_2_ (P90)	TiO_2_ (rutile)	ZnO	NiO	CeO_2_	SiO_2_
SP-D in BALF (ng/mL)	Control	1644 ± 235	874 ± 79	997 ± 127	667 ± 271	754 ± 103	531 ± 126
	0.2 mg	2299 ± 394	1084 ± 349	986 ± 35	1519 ± 493	1440 ± 267	-
	1.0 mg	2037 ± 632	1317 ± 717	829 ± 76	4852 ± 1036	3010 ± 61	2194 ± 253
Exposure/control ratio of BALF	0.2 mg	1.4	1.2	1.0	2.3	1.9	-
	1.0 mg	1.2	1.5	0.8	7.3	4.0	4.1
SP-D in serum (ng/mL)	Control	84 ± 30	-	-	84 ± 6	-	112 ± 9
	0.2 mg	73 ± 8	-	-	127 ± 59	-	-
	1.0 mg	85 ± 16	-	-	136 ± 18	-	136 ± 8
Exposure/control ratio of serum	0.2 mg	0.9	-	-	1.5	-	-
	1.0 mg	1.0	-	-	1.6	-	1.2

**Table 2 ijms-22-04635-t002:** Result of ROC analysis in intratracheal instillation of nanomaterials.

	Time	AUC	95% CI	*p* Values
SP-D in BALF	3 days	0.463	0.296–0.631	0.663
	1 week	0.803	0.674–0.933	0.000
	1 month	0.982	0.952–1.000	0.000
	3 months	0.795	0.670–0.920	0.000
	6 months	0.717	0.567–0.866	0.010
SP-D in serum	3 days	0.424	0.144–0.703	0.597
	1 week	0.875	0.694–1.000	0.009
	1 month	1.000	1.000–1.000	0.001
	3 months	0.903	0.750–1.000	0.005
	6 months	0.521	0.232–0.810	0.885

**Table 3 ijms-22-04635-t003:** Characterization of inhaled chemicals including nanomaterials.

Samples	Toxicity	Characterization	Animal (Rat)	Dose	Lung Inflammation	Reference
NiO	High	Size 19 nm, BET 57 m^2^/g Secondary particle diameter (DLS) 59.7 nm	Male Fischer 344	0.2 mg/rat, 1.0 mg/rat	+	[25]
CeO_2_	High	Size 7.8 nm, BET 101 m^2^/g Secondary particle diameter (DLS) 10.0 nm	Male Fischer 344	0.2 mg/rat, 1.0 mg/rat	+	[26]
TiO_2_ (P90)	Low	Size 14 nm, BET 104 m^2^/g Secondary particle diameter (DLS) 22.7 nm	Male Wistar Hannover	0.2 mg/rat, 1.0 mg/rat	−	[28]
TiO_2_ (rutile)	Low	Size 12 nm × 55 nm, BET 111 m^2^/g Secondary particle diameter (DLS) 44.9 nm	Male Fischer 344	0.2 mg/rat, 1.0 mg/rat	−	[25]
ZnO	Low	Size 35 nm, BET 31 m^2^/g Secondary particle diameter (DLS) 33 nm	Male Fischer 344	0.2 mg/rat, 1.0 mg/rat	−	[27]
SiO_2_	High	Primary particle size 1.6 µm	Male Fischer 344	1.0 mg/rat	+	[29]

Lung inflammation +: persistent inflammation, −: transient inflammation.

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
