# Peer review of "Examination of Surfactant Protein D as a Biomarker for Evaluating Pulmonary Toxicity of Nanomaterials in Rat"

_ijms, 2021, doi:10.3390/ijms22094635_

Round 1

Reviewer 1 Report

The authors have convincingly demonstrated the value of surfactant protein-D (SP-D) as a biomarker of nanaparticle induced pulmonary toxicity in rat, by comparing its presense  in  bronchoalveolar-lavage fluid (BALF) and serum after intrtrachral exposure to toxic nanoparticles (NaO and CeO2) to that observed after exposure to nontoxic ZnO nanoparticles. The results were further confirmed by measuring the cellular expression of SP-D in treated rats as well as with immunohistochemical methods. 

While I have no comments regarding the general outline of the study and its general conclusions, I wonder whether the dosing of the rats might be expressed differently. Two dose levels of nanoparticles were used (0.2 and 1. 0 mg per rat). I wonder, whether it might be possible to express the dosing also as the actual number of nanoparticles per animal. The toxicity of nanoparticles  is probably more related to the size and number of the particles than to their actual weight.   If the actual number of particles turns out to be significantly different between the treatment groups, the results are not invalidated, but the finding should be discussed. 

Reviewer 2 Report

The manuscript is easy to follow however I have specific concerns outlined below.

Results:

  1. In Fig 1, Stainings at time 0 point are missing which are important controls. Can authors quantify the infiltration of neutrophils and macrophages using FACS? It would further strengthen the results. An indication in staining images with arrows would be helpful to readers. The authors should justify the dose selection, why 1mg was used? Why distilled water was used instead of PBS for negative control. Do the authors have a positive control of nanomaterials which is known to cause a significant lung toxicity?
  2. In Fig 2, Why SiO2 0.2mg was not shown? At low dose 0.2mg, nickel and cerium showed higher SP-D concentrations in BALF while at high dose 1 mg, cerium and zinc showed higher SP-D concentrations in BALF. In both cases, there is pattern indicating SP-D concentrations peak at 1 week but then levels off at 3 months and 6 months. Can authors explain this? Why different nanomaterials should differentially regulate the SP-D concentrations? Further, it would be helpful to add time 0 of SP-D concentrations in BALF. The negative control bar could be taken as an average and used a single bar rather than repeating and then perform statistics for better representation of graphs.
  3. The authors should abbreviate HO in text. The authors should introduce inflammatory markers in the Introduction. The authors should write detailed results for Fig 3. Do the authors have ELISA for macrophage in BALF? What is rationale to measure rat HO-1 and activity of LDH?

  1. In Fig 4, negative control bar could be averaged and used as beginning as a control and then compare with other groups. It should not change after 6 months. Nickel, cerium and also zinc shows higher SP-D expression at 0.2mg. at 1mg, although almost all nanomaterials exhibited higher SP-D expression between 3 days and 1 week however more apparent is nickel, cerium and silica. Can authors explain why expression of SP-D is differentially regulated? Again time 0 is missing.

  1. In Fig 5, please indicate with arrows what should be looked for? Why Nickel was chosen at 3 months? How does staining look at 6 months? Is expression higher or levelled off? Hwy cerium and silica was eliminated?

  1. In Fig 6, remove comma from line 127. Again negative controls should be averaged and used as one bar. Why authors focused on nickel only? How about cerium? It looks like SP-D concentration levels off for nickel at 6 months. Can authors explain this result?

  1. Although a slight correlation of SP-D concentration between BALF and serum was observed, it would be helpful to see the inflammation correlation with respect to inflammatory markers investigated.

  1. In Fig 8, ROC curves for SP-D concentration showed around 1. Can author further elaborate these results and discuss in the discussion?

Discussion:

  1. Why serum and BALF should show differential regulation of SP-D concentrations when exposed to nanomaterials?
  2. Can authors show evidence of sufficient inflammation at each time point?
  3. Why Male Fisher rats and male wistar rats were used or different nanomaterial exposure? Why only males were investigated?
  4. How authors confirm nanomaterial deposition in lung?

Round 2

Reviewer 2 Report

The manuscript quality is greatly improved.